# Monitoring Radiotherapeutic Response in Prostate Cancer Patients Using High Throughput FTIR Spectroscopy of Liquid Biopsies

**DOI:** 10.3390/cancers11070925

**Published:** 2019-07-02

**Authors:** Dinesh K.R. Medipally, Thi Nguyet Que Nguyen, Jane Bryant, Valérie Untereiner, Ganesh D. Sockalingum, Daniel Cullen, Emma Noone, Shirley Bradshaw, Marie Finn, Mary Dunne, Aoife M. Shannon, John Armstrong, Fiona M. Lyng, Aidan D. Meade

**Affiliations:** 1Radiation and Environmental Science Centre, Focas Research Institute, Technological University Dublin, D08 NF82 Dublin, Ireland; 2School of Physics & Clinical & Optometric Sciences, Technological University Dublin, D08 NF82 Dublin, Ireland; 3BioSpecT EA 7506, Université de Reims Champagne−Ardenne, UFR Pharmacie, 51097 Reims, France; 4Plateforme en Imagerie Cellulaire et Tissulaire (PICT), Université de Reims Champagne−Ardenne, 51097 Reims, France; 5Clinical Trials Unit, St Luke’s Radiation Oncology Network, St Luke’s Hospital, D06 HH36 Dublin, Ireland; 6Cancer Trials Ireland, D11 KXN4 Dublin, Ireland; 7Department of Radiation Oncology, St Luke’s Radiation Oncology Network, St Luke’s Hospital, D06 HH36 Dublin, Ireland

**Keywords:** radiotherapy, toxicity, prostate cancer, blood plasma, high throughput, Fourier transform infrared spectroscopy

## Abstract

Radiation therapy (RT) is used to treat approximately 50% of all cancer patients. However, RT causes a wide range of adverse late effects that can affect a patient’s quality of life. There are currently no predictive assays in clinical use to identify patients at risk of normal tissue radiation toxicity. This study aimed to investigate the potential of Fourier transform infrared (FTIR) spectroscopy for monitoring radiotherapeutic response. Blood plasma was acquired from 53 prostate cancer patients at five different time points: prior to treatment, after hormone treatment, at the end of radiotherapy, two months post radiotherapy and eight months post radiotherapy. FTIR spectra were recorded from plasma samples at all time points and the data was analysed using MATLAB software. Discrimination was observed between spectra recorded at baseline versus follow up time points, as well as between spectra from patients showing minimal and severe acute and late toxicity using principal component analysis. A partial least squares discriminant analysis model achieved sensitivity and specificity rates ranging from 80% to 99%. This technology may have potential to monitor radiotherapeutic response in prostate cancer patients using non-invasive blood plasma samples and could lead to individualised patient radiotherapy.

## 1. Introduction

Prostate cancer is the second most frequently diagnosed cancer and the third most common cause of death from cancer in men [1]. Treatment options for prostate cancer are based on factors including the patient’s life expectancy, tumour characteristics including tumour stage, tumour aggressiveness and prostate specific antigen (PSA) level. Generally, low-risk prostate cancer patients are treated with radical prostatectomy, low-dose-rate brachytherapy and active surveillance, and high-risk prostate cancer patients are treated with a combination of hormone therapy and radiotherapy [2]. In the high-risk group of patients, combined treatment with hormone therapy and radiotherapy (RT) produces a therapeutic gain [3]. However, this combination of treatment also results in serious side effects which may affect the patient’s quality of life. These side effects include acute and late radiation toxicity. Acute side effects develop during the treatment phase and generally last for a few weeks after treatment is completed. Late side effects begin months to years after radiation therapy, are generally irreversible and can continue for several years after completion. Late toxicities of radiation treatment in prostate cancer patients include sexual dysfunction, persistent bowel problems (such as intermittent rectal bleeding, tenesmus and loose stools) and urinary obstructive symptoms [4]. The known causes of radiation toxicity include radiation dose, diabetes, concurrent chemotherapy and intrinsic radiosensitivity [5]. However, the radiation response varies from patient to patient, and to date no markers are in clinical use for the prediction of normal tissue toxicity or treatment outcome.

The development of predictive assays for normal tissue radiosensitivity has been a focus of research for several years and to date no assay has been translated into the clinical setting. In this study, radiosensitivity was not measured directly. Rather, the patient’s response to radiation treatment was measured, though these are related. The first gold-standard in vitro technique for measuring radiosensitivity was the clonogenic assay. The clonogenic assay determines the ability of a cell line to multiply indefinitely, in a way that retains its reproductive ability by forming a colony [6]. This assay is generally labour intensive and time consuming because the formation of colonies can take several weeks to months [7]. Streffer et al. developed a test for assessing radiosensitivity based on micronuclei formation after irradiation [8]. Micronuclei formation assays measure the ratio of the number of S phase cells to the total number of cells with micronuclei. The limitation of this assay is that the formation of micronuclei varies for different cell types due to the different tolerances for genetic loss [9]. The G2 chromosomal assay is one of the most used predictive cytogenetic based radiosensitivity assays. The assay is a positive predictor of radiosensitivity and corresponds well with cancer predisposition [10]. The G2 chromosomal radiosensitivity assay can be performed on patient blood samples and has been shown to be reliable for predicting patient radiosensitivity. This assay is relatively time consuming with some observer subjectivity. DNA damage assays have been used to predict radiosensitivity. The most commonly used DNA damage assay is the γH2AX foci assay which gives a quick read-out of DNA damage. Although Bourton et al. showed excellent potential for distinguishing patients with toxicity compared to normal patient reactions, the γH2AX assay was not sensitive enough to distinguish patients with extreme sensitivity or patients with a moderate severity of toxicity [11].

More recently, several radiogenomics studies have reported the development of radiotoxicity due to single nucleotide polymorphisms (SNPs) in candidate genes such as TGFB1, XRCC3, XRCC1, ATM [12] results which have been replicated in recent large-scale studies [13,14]. The preliminary results of genome wide association studies (GWAS) show evidence of an association between common genetic variants and a patient’s risk of developing radiation toxicity [15,16,17]. GWAS have successfully identified several common, modest risk variants for many widespread complex diseases [18,19,20,21,22] but have been unsuccessful for other phenotypes. The main limitations of genomic assays are their relatively high cost and that they are labour intensive. 

The present study proposes a novel approach based on FTIR spectroscopy for the detection of radiation induced toxicity using liquid biopsies. FTIR spectroscopy is based on the interaction of infrared light with a molecular species, providing a non-destructive, label-free method for studying molecular composition and structure either in isolation or within complex biological systems such as cells, tissues and biofluids. FTIR spectroscopy has proved its potential to detect and classify various diseases and cancers including galactosemia [23], Alzheimer’s disease [24], hepatic fibrosis [25], ovarian cancer [26], gastrointestinal cancers [27] and breast cancer [28] using blood plasma and serum samples. Recently, FTIR spectroscopy has also been applied to radiobiological analysis. FTIR spectroscopy has been demonstrated to be sensitive to molecular events occurring in cells and tissue after exposure to ionising radiation. These applications include retrospective biological dosimetry [29], the analysis of both targeted and non-targeted effects of ionising radiation [30] and the prediction of DNA damage levels after in vitro irradiation [31].

In the present study, infrared spectra were acquired from plasma samples obtained from patients before radiotherapy (baseline) and at follow up. Changes in spectral profiles as a result of hormone treatment and radiotherapy in prostate cancer patients were investigated. Significant spectral differences were observed between the plasma samples of patients at baseline, post hormone therapy, post radiotherapy and at two and eight-month follow ups. Similarly, significant spectral differences were observed in the patients showing minimal and severe acute and late toxicity. The acquired infrared spectra were also analysed by principal component analysis (PCA) and partial least squares discriminant analysis (PLS-DA). The PLS-DA classifier was able to discriminate these patient groups with sensitivity and specificity rates ranging from 80% to 99%. This is the first study which explores the feasibility of FTIR spectroscopy for monitoring radiotherapeutic response in prostate cancer patients using minimally invasive blood plasma samples.

## 2. Results

### 2.1. Monitoring Treatment Progression

#### 2.1.1. Changes in the Spectral Features with Patient Treatment Progression

Figure 1 shows the comparison of mean plasma spectra of different patient groups. To explain the differences between each group, difference spectra were computed by subtracting the mean spectra of plasma from the patients at different treatment time points from their mean spectra at baseline (Figure 2). Similarly, to explain the differences between the plasma from patients post hormone treatment and post radiotherapy, difference spectra were computed by subtracting the mean spectra of plasma from the patients at different time points post radiotherapy from their mean spectra post hormone treatment (Figure 3). Statistically significant differences (*p* < 0.001) were observed between the patient groups. Differences in the form of intensity related variations were observed across these mean spectra: in the regions around 1024 and 1050 cm^−1^ (C−O stretching and bending vibrations of glycogen) [32], 1070–1090 cm^−1^ (symmetric stretching of PO_2_^−^, nucleic acids, phospholipids and saccharides) [33], 1120–1170 cm^−1^ (stretching vibrations of (C−O) and ν(C−O−C) of carbohydrates) [23], 1230–1330 cm^−1^ (amide III) [34], 1400 cm^−1^ (stretching vibrations of COO^−^, amino acids) [23], 1450 cm^−1^ (CH_2_ scissoring, proteins) [33], 1545 cm^−1^ (amide II) [33], 1656 cm^−1^ (amide I) [33], 1740–1760 cm^−1^ (stretching vibrations of (C=O) of fatty acids, triglycerides and cholesterol esters) [33], 2800–2960 cm^−1^ (stretching vibrations of (CH_2_/CH_3_) of lipids, fatty acids, triglycerides and proteins) [33], 3300 cm^−1^ (amide A) [23], and 3400–3600 cm^−1^ (hydroxyl (OH) stretch of carboxylic acids) [35] between patients at baseline and patients at other treatment stages.

#### 2.1.2. PCA

PCA was performed on second derivative mean spectra using the wavenumber range of 3500−1000 cm^−1^. For clarity, only the mean spectra are represented for each patient. PCA was carried out with 10 principal components (PCs) in which the first four PCs accounted for ∼80% of the total percentage. The first two PCs were used to visualise the classification between the groups (Figure 4A).

#### 2.1.3. Classification of Plasma Spectra of Patients during Treatment with Respect to Baseline by PLS-DA

Classifications of plasma spectra from patients at baseline against various treatment stages were performed using PLS-DA to investigate whether the patient groups could be discriminated based on the observed spectral features. The classification sensitivities and specificities are provided in Table 1 and are calculated for the cross validated PLS-DA model. An example of the cross-validation performance of the PLS-DA model is shown in Figure 5.

### 2.2. Analysis of Patient Toxicity after Radiotherapy

#### 2.2.1. Changes in the Spectral Features with the Onset of Acute and Late Toxicity

All patients suffered from either acute grade 1 (mild) or grade 2 or higher (severe) toxicities immediately following completion of radiotherapy. A grade 0 level to toxicity implies no toxicity. At eight months post radiotherapy toxicity in some patients was resolved, while in others late toxicity developed for the first time. Table 2 shows the number of prostate cancer patients showing acute toxicity immediately following completion of radiotherapy and late toxicity at eight months post radiotherapy. 

In this study, difference spectra between those measured pre-therapy at baseline and those measured immediately after completion of radiotherapy and at 8 months follow up from patients with grade 0−1 and grade 2+ acute and late toxicity were calculated to examine the changes in spectral features at various toxicity levels (Figure 6). Major changes were observed in the regions around 1020−1090 cm^−1^ (glycogen, nucleic acids), 1330−1430 cm^−1^ (amide III), 1545 cm^−1^ (amide II), 1656 cm^−1^ (amide I), 1740−1760 cm^−1^ (fatty acids, triglycerides and cholesterol esters), 2800−2950 cm^−1^ (lipids, triglycerides, fatty acids and proteins), 3300 cm^−1^ (amide A) and 3400−3600 cm^−1^ (OH stretch) between the plasma spectra from patients showing grade 0−1 and grade 2+ acute and late toxicity.

#### 2.2.2. PCA

PCA was performed on second derivative mean spectra with the same conditions as mentioned earlier. Figure 7 shows the scatter plot and loading plot of the PCA classification of acute grade 0–1 and acute grade 2+ toxicity plasma spectra. Figure 8 shows the scatter plot and loading plot of the PCA classification of late grade 0–1 and late grade 2+ toxicity plasma spectra. In both cases, the first four PCs accounted for ∼93% of the total percentage variance and the first two PCs were used to visualise the classification between the groups.

#### 2.2.3. PLS−DA

Classifications of plasma spectra from patients showing acute and late toxicity were performed to investigate if the two patient groups could be discriminated. PLS-DA models were developed as described earlier. The resulting sensitivities and specificities are provided in Table 3. 

## 3. Discussion

In the first part of the study, FTIR spectra acquired from blood plasma samples from patients at baseline, post hormone, post radiotherapy and at follow up were used to monitor the changes with treatment progression (Figure 2). The following hormone therapy increases were observed in the regions around 1000−1090 cm^−1^, 1120−1170 cm^−1^, 1230−1330 cm^−1^, 1520 cm^−1^, 1590 cm^−1^, 1656 cm^−1^, 1740−1760 cm^−1^, 2800−2960 cm^−1^ and 3300 cm^−1^, and decreases were observed in the region around 3400−3600 cm^−1^ in the spectra of plasma. The increase in the regions around 1230−1330 cm^−1^, 1520 cm^−1^, 1590 cm^−1^, 1656 cm^−1^ and 3300 cm^−1^ suggests an increase in the protein levels after hormone therapy. The increased 1740−1760 cm^−1^ and 2800−2960 cm^−1^ regions indicate the elevated lipid profile after hormone therapy. The increase in lipid content can occur as a result of a decrease in testosterone levels after androgen deprivation therapy [36]. The increased levels of triglycerides and total cholesterol in the serum of prostate cancer patients after androgen deprivation therapy have been reported previously [37]. The increases in the regions around 1020−1090 cm^−1^ and 1120−1170 cm^−1^ are associated with saccharides and carbohydrates. Increased levels of HbAlc (hemoglobin bound to glucose) in blood has been previously observed in prostate cancer patients treated with androgen deprivation therapy [38].

The increase in protein and lipid content observed after hormone therapy resolved after the course of radiotherapy. Decreases in the regions from 1230−1330 cm^−1^, 1545 cm^−1^, 1656 cm^−1^, 1740−1760 cm ^−1^, 2800−2960 cm^−1^ and 3300 cm^−1^ suggests a decrease in protein, lipid and fatty acid content. A decline in lipid concentration in the serum of prostate cancer patients after radiotherapy has been reported previously [39]. The increased vibrations in the 1000−1100 cm^−1^ region suggests an increase in glycogen and nucleic acids after radiotherapy. The increase in nucleic acids after radiotherapeutic treatment suggests an increase in the levels of circulating DNA. The elevated plasma levels of tumoural cell free DNA was observed in non-small cell lung cancer patients after radiotherapy [40]. The decreased OH stretch band (3400–3600 cm^−1^) observed after hormone therapy was increased after radiotherapy. 

The decrease in fatty acid (1740–1760 cm^−1^) and lipid content (2800–2960 cm^−1^) observed after radiation therapy increased after two and eight months RT. Jelonek et al. [41] reported a significant decrease in serum lipids immediately after radiotherapy and then an increase in levels of lipids in serum samples at follow up. The decrease in the vibrations at 1230–1330 cm^−1^, 1545 cm^−1^, 1656 cm^−1^ and 3300 cm^−1^ and increase in the vibrations at 3400–3600 cm^−1^ at eight months post therapy are consistent with the changes observed immediately following radiotherapy. The decreased vibrations in the regions 1020–1090 cm^−1^ were also observed in the plasma spectra of patients at two and eight months post RT.

Similar spectral differences were observed when plasma from patients post hormone treatment and from patients post radiotherapy (Figure 3) were compared. However, decreases in glycogen, nucleic acid (1000–1100 cm^−1^) and lipid content (2800–2960 cm^−1^) were observed in the patients post radiotherapy and at follow up compared to the patients post hormone treatment. 

The PCA analysis showed minimally overlapping clusters between the analysed groups (Figure 4A). PC-1 explains of 51.7% of total variance. The PC-1 loading plot shows positive bands at 1560 cm^−1^ and 1656 cm^−1^ and negative bands at 1747 cm^−1^, 2854 cm^−1^, 2873 cm^−1^, 2930 cm^−1^, 2960 cm^−1^ and 3013 cm^−1^. The positive bands in PC-1 are assigned to proteins and the negative bands are assigned to lipids, fatty acids and triglycerides. PC-2 explains 15.6% of total variance and the PC-2 loading plot shows positive bands related to proteins and lipids (1560 cm^−1^, 1634 cm^−1^, 1644 cm^−1^, 2851 cm^−1^ and 2870 cm^−1^) and negative bands related lipid and fatty acids (2946 cm^−1^). The PCA analysis confirms that protein, lipids, fatty acids and triglycerides are the main discriminating features in spectra from plasma of patients at baseline and those at follow up time points.

The optimised PLS-DA model was able to discriminate baseline/post hormone therapy, baseline/post radiotherapy, baseline/two months after RT and baseline/eight months after RT with sensitivity and specificity rates ranging from 80% to 99%. The classification rate increased with the progression of treatment (Table 1) and to explain this classification trend, prostate specific antigen (PSA) levels at baseline and at each treatment time point were compared (Table 4). The PSA levels decreased significantly after hormone therapy and a slight decrease in PSA level was observed after radiotherapy. The PSA levels at follow up were similar to those observed immediately after radiotherapy. Thus, the classification trend did not correlate with the observed PSA levels. The increase in the classification with treatment progression may be associated with the development of acute and late toxicity after radiotherapy.

In the second part of the study, FTIR spectra of plasma from prostate cancer patients suffering from acute and late toxicity after radiotherapy were used to monitor the changes with the progression of toxicity (Figure 6). For patients showing acute grade 0−1 toxicity, an increase in glycogen, nucleic acids (1020–1090 cm^−1^) and OH stretch (3400–3600 cm^−1^), and a decrease in amide III (1330–1430 cm^−1^), amide II (1545 cm^−1^), amide I (1656 cm^−1^), fatty acids/triglycerides (1740–1760 cm^−1^), amide A (3300 cm^−1^) and lipids/proteins (2800–2950 cm^−1^) were observed. Patients suffering with acute grade 2+ toxicity showed an increase in glycogen and nucleic acid bands (1020–1090 cm^−1^) compared to the patients suffering with acute grade 0−1 toxicity. Similar increases were also observed in patients with late grade 2+ toxicity compared to patients with late grade 0–1 toxicity. In both acute and late toxicity, a decrease in protein bands (1330–1430 cm^−1^, 1545 cm^−1^, 1656 cm^−1^ and 3300 cm^−1^) and an increase in the OH stretch band was observed. The decrease in the protein content might be due to the oxidation of proteins after RT. Oxidation of protein induced by free radicals produced by RT is a major cause of healthy tissue damage [42]. In patients showing late grade 0−1 and grade 2+ toxicity, an increase in fatty acids/triglycerides (1740–1760 cm^−1^) and lipids (2800–2960 cm^−1^) was observed compared to patients showing acute toxicity and this increase was significant in patients showing late grade 2+ toxicity. This may suggest that an increase in lipids and fatty acids/triglycerides contributes to the development of late toxicity. An association between alterations in the serum lipidome and the development of treatment toxicity has been previously observed [41].

The PCA analysis showed clear discrimination between the plasma spectra from patients showing acute grade 0–1 and grade 2+ toxicity (Figure 7A). PC-1 and PC-2 explain 70.2% and 14.7% of the total variance respectively. PC-1 showed positive bands at 1040 cm^−1^, 1067 cm^−1^, 1090 cm^−1^, 1120 cm^−1^, 1640 cm^−1^, 1691 cm^−1^, 2853 cm^−1^, 2874 cm^−1^ and 2963 cm^−1^, and a negative band at 1659 cm^−1^. PC-2 showed positive bands at 1040 cm^−1^, 1067 cm^−1^, 1090 cm^−1^, 1120 cm^−1^, 1418 cm^−1^, 1656 cm^−1^, 1736 cm^−1^, 2853 cm^−1^ and 2926 cm^−1^, and a negative band at 1632 cm^−1^. The loading plot reveals that the prominent discriminating features between the plasma spectra from patients showing acute grade 0–1 and grade 2+ toxicity are associated with saccharides, DNA, RNA (1040 cm^−1^, 1067 cm^−1^, 1090 cm^−1^ and 1123 cm^−1^), amide I (1640 cm^−1^, 1656 cm^−1^ and 1691 cm^−1^), fatty acids, triglycerides and lipids (1736 cm^−1^, 2874 cm^−1^, 2853 cm^−1^, 2926 cm^−1^ and 2963 cm^−1^).

The PCA analysis showed minimally overlapped clusters between the plasma spectra from patients showing late grade 0−1 and late grade 2+ toxicity (Figure 8A). PC-1 explains 65% of total variance and showed positive bands at 1032 cm^−1^, 1080 cm^−1^, 1516 cm^−1^, 1632m^−1^, 1747 cm^−1^, 2853 cm^−1^, 2874 cm^−1^ and 2963 cm^−1^, and negative bands at 1545 cm^−1^ and 1655 cm^−1^. PC-2 explains 16% of total variance and showed positive bands at 1042 cm^−1^, 1067 cm^−1^, 1090 cm^−1^, 1123m^−1^, 1640 cm^−1^, 2855 cm^−1^ and 2926 cm^−1^, and negative bands at 1516 cm^−1^, 1550 cm^−1^ and 1624 cm^−1^. The loading plot reveals that the major discriminating features between the plasma spectra from patients showing late grade 0−1 and grade 2+ toxicity includes the vibrations of saccharides, DNA, RNA (1032 cm^−1^, 1067 cm^−1^, 1080 cm^−1^, 1090 cm^−1^ and 1123 cm^−1^), amide I (1624 cm^−1^, 1632 cm^−1^, 1640 cm^−1^ and 1655 cm^−1^), amide II (1516 cm^−1^, 1545 cm^−1^ and 1550 cm^−1^), fatty acids, triglycerides and lipids (1747 cm^−1^, 2855 cm^−1^, 2874 cm^−1^, 2926 cm^−1^ and 2963 cm^−1^).

The PLS-DA model classified plasma spectra from patients showing grade 0–1 and grade 2+ acute toxicity with a sensitivity and specificity of 80.8% and 81.6%, respectively. The model classified plasma spectra from patients showing grade 0–1 and grade 2+ late toxicity with a sensitivity and specificity of 81.4% and 81.5%, respectively.

However, it is acknowledged that this is a small study cohort and a larger sample size would be required to validate the findings. In addition, all patients in the study received hormone therapy and there was no radiotherapy alone group. Although our study focused on gastrointestinal (GI) and genitourinary (GU) toxicity and did not include specific hormonal toxicities, further work should include patients treated with radiotherapy only. Future work should also involve biological assays showing protein, lipid, fatty acid and circulating DNA levels to corroborate these findings.

## 4. Materials and Methods

### 4.1. Ethical Approval

Blood samples were obtained from prostate cancer patients enrolled in a trial detailed in the section following.

The translational research study was approved by the St Luke’s Radiation Oncology Network Research Ethics Committee and all research was performed in accordance with relevant guidelines and regulations. Informed consent was obtained from all participants. Fresh whole blood was drawn into Li-heparin tubes at St. Luke‘s Radiation Oncology Network, St. Luke’s Hospital (SLRON SLH), in Dublin, and were coded before being transferred to the Technological University (TU) Dublin laboratory.

### 4.2. Patients

The prostate cancer patients for this study were recruited from the Cancer Trials Ireland (formerly All Ireland Cooperative Oncology Research Group, ICORG) trial 08−17 which is entitled “A Prospective Phase II Dose Escalation Study Using intensity modulated radiotherapy (IMRT) for High Risk N0 M0 Prostate Cancer (NCT00951535)”. The primary endpoint is to determine if dose escalation up to 81 Gy using IMRT for high risk localised prostate cancer can provide prostate specific antigen (PSA) relapse-free survival similar to that previously reported [43]. All patients were prescribed either six months or three years of neo-adjuvant/adjuvant hormone therapy using non−steroidal anti androgens (NSAA) and luteinizing hormone releasing hormone (LHRH). After the neoadjuvant hormone therapy, patients were treated with radiation therapy, where dose escalation was allowed up to a maximum of 81 Gy from a baseline 75.6 Gy, with treatment delivered by intensity modulated radiotherapy. PSA levels and gastrointestinal and genitourinary (GI/GU) toxicities were recorded prior to treatment, during treatment and at follow up using the National Cancer Institute Common Terminology Criteria for Adverse Events (NCI−CTCAE) grading system, version 3. Patients are followed up regularly at two months post radiation therapy (RT), eight months post RT, six-monthly thereafter until Year 5, and annually thereafter until Year 9. For this translational study, a total of 53 prostate cancer patients were enrolled and of these patients, plasma samples were collected from patients at baseline (*n* = 37), post hormone therapy (*n* = 36), post radiotherapy (*n* = 43), two months after RT (*n* = 37) and eight months after RT (*n* = 35). A summary of clinical features of the prostate cancer patients is given in Table 5.

### 4.3. Plasma Separation

Whole blood was drawn into Lithium−heparin tubes. Plasma was isolated from these blood samples by centrifugation at 3600 g for 5 min at 18 °C. The samples were subsequently stored at −80 °C prior to FTIR acquisition.

### 4.4. FTIR Spectroscopy

Plasma samples stored at −80° C were thawed at room temperature and were diluted threefold in physiological water. A volume of 4 μL diluted plasma sample was deposited on a 384−well silicon plate (Bruker Optics GmbH, Ettlingen, Germany), and air-dried at room temperature. For each sample, 10 spots were used giving 10 instrumental replicates. The plate was then inserted into a high-throughput module (HTS−XT, Bruker Optics GmbH) attached to an FTIR spectrometer (Tensor 27, Bruker Optics GmbH). FTIR spectra were acquired in the transmission mode using the OPUS v6.5 software (Bruker Optics GmbH) in the wavenumber range from 4000 to 400 cm^−1^, using a spectral resolution of 4 cm^−1^ and 32 co-additions. FTIR spectra were then subjected to a quality test (OPUS v6.5) and details of this test are fully described in reference [23,44]. Spectra that passed the quality test were pre-processed and processed in the wavenumber range from 800 to 4000 cm^−1^. A summary of the high throughput (HT)-FTIR methodology for analysis of blood plasma samples is presented in Figure 9.

### 4.5. Data Analysis

#### 4.5.1. Pre-Processing

Pre-processing of raw spectra is important to enhance the accuracy and stability of spectra for later analysis [45]. Pre-processing includes baseline correction, calculation of second derivative spectra and vector normalisation. Baseline correction was performed using the rubberband baseline subtraction [46]. Second derivative spectra were calculated using the Savitzky–Golay algorithm [47] and a window length of night points. Second derivative spectra allow more distinct identification of small and adjacent lying absorption peaks which are not clearly discernable in the original spectrum. Prior to analysis, all spectra were standardized using vector normalization.

#### 4.5.2. PCA

PCA is a form of unsupervised multivariate analysis, which has become a standard processing technique for Raman and FTIR spectral data. Mainly, PCA is a feature selection process that enables the user to identify variances in the data set that may be used to classify objects into certain groups. PCA explains the variance in the data by finding combinations of the original dimensions that describe the largest variance between the data sets where the output is in the form of a complementary set of scores and loading plots termed principal components (PCs) [48]. PCA is performed by subtracting the mean of the data set to obtain a mean centered matrix, calculating the covariance matrix and then the eigenvectors and eigenvalues of the covariance matrix. The eigenvector with the largest eigenvalue describes the largest source of variance across all the spectra. The eigenvector with the largest source of variance is also called the first principal component [49]. The second principal component is the eigenvector with the next largest eigenvalue and describes the second largest source of variance. The results are presented using the scores of the most explained PCs. Since PCA is a tool for data visualisation and trends, subsequent PLS-DA was also carried out for classification of plasma spectra of different patient groups. 

#### 4.5.3. PLS-DA

PLS-DA is one of the generally applied classification technique in chemometrics [50]. PLS-DA is a discriminant analysis approach derived from partial least squares regression [51] and involves the production of regression model between the data matrix and the discrete spectral class [50]. The aim of PLS-DA is to achieve maximum covariance between the predictor variables (i.e., FTIR spectra) and response variables (i.e., class label of corresponding spectra) of a multidimensional dataset by determining a feature subspace that differentiates the multidimensional feature space into two discrete regions, with as negligible an error as achievable. This new feature subspace enables the prediction of dependent variables using fewer factors referred to as latent variables (LVs) [52]. These factors explain the behavior of the response variables and populate the new feature subspace onto which the predictor variables are superimposed [52]. PLS-DA gives various statistics like loading weight and regression coefficient which can be employed for the identification of most important variables [53,54]. It also gives a visual interpretation of complicated datasets through an easily illustratable scores plot that explains the separation between different groups [55].

PLS-DA classification was applied on the pre-processed second derivative FTIR spectra of plasma collected from prostate cancer patient groups. To find the optimal number of LVs, PLS-DA was executed with a variable number of LVs, 1≤m≤50. For each m LV(s), 20 rounds of PLS-DA cross validation were performed. The PLS-DA model at each round was built and trained using the spectra of 90% of patients, which were randomly selected from each patient group. The remaining spectra of 10% of the patients were used for validating the trained model. The sensitivities and specificities of each PLS-DA of m LVs were then averaged through 20 rounds of PLS-DA applications.

## 5. Conclusions

This is the first biofluid based FTIR spectroscopy study that explores the discrimination between blood plasma at various radiotherapeutic treatment stages, as well as the discrimination between plasma spectra with treatment outcome, here measured as treatment toxicity. Spectral discrimination was demonstrated between infrared spectra of plasma samples from prostate cancer patients at each treatment stage and follow-up time point, as well as between spectra from patients showing both acute and late toxicity and toxicity grade. The PLS-DA model classified the patient groups with high sensitivity and specificity. Biofluid based FTIR spectroscopy is reagent free, label free, cost effective and rapid; this research has demonstrated that it may also provide a useful adjunct to radiotherapy treatment planning and monitoring within the clinical workflow.

## Figures and Tables

**Figure 1 cancers-11-00925-f001:**
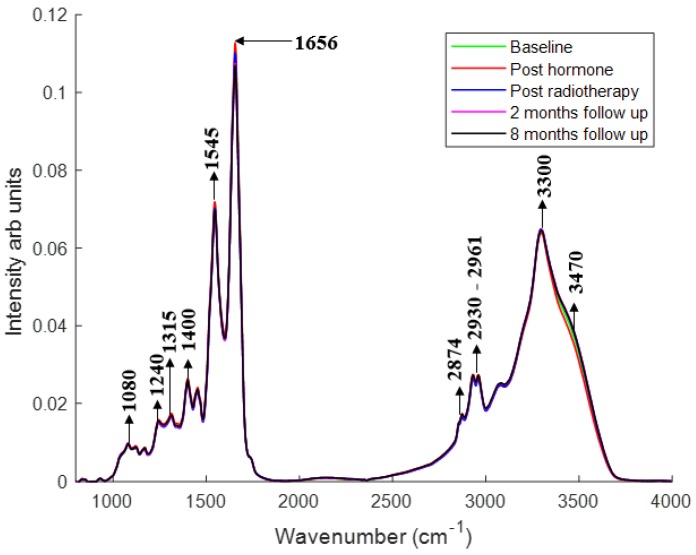
Mean FTIR spectra of plasma from patients at baseline, post hormone treatment, post radiotherapy, two and eight months after radiotherapy. Spectra were baseline corrected and vector normalized. Major bands are highlighted.

**Figure 2 cancers-11-00925-f002:**
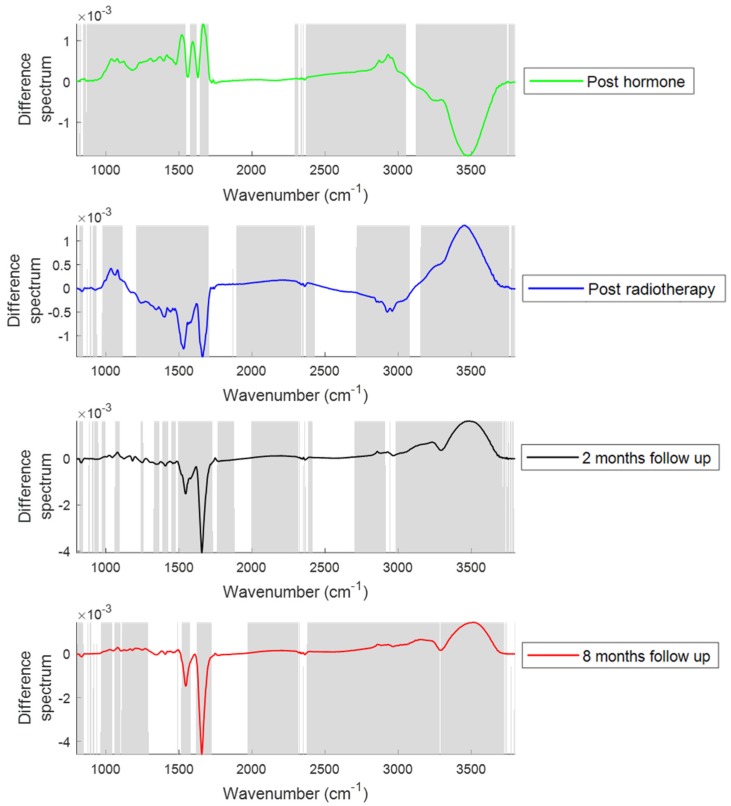
Difference spectra of plasma from patients post hormone treatment, post radiotherapy, at two months follow up and at eight months follow up. Difference spectra were computed by subtracting the mean spectra of plasma from the patients at different treatment time points from their mean spectra at baseline. The shaded regions depict the spectral regions which are significantly different between each sample set using a two-tailed *t*-test with *p* < 0.001.

**Figure 3 cancers-11-00925-f003:**
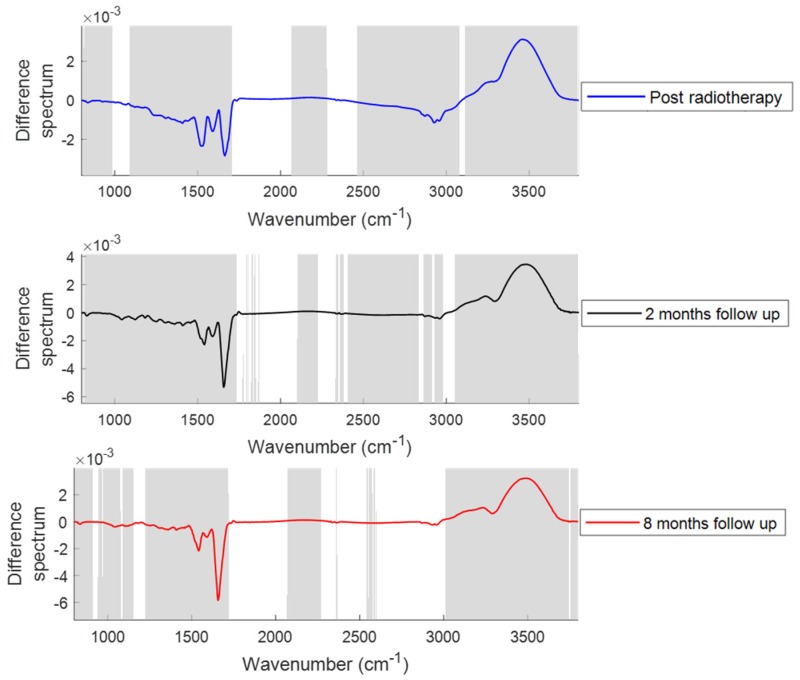
Difference spectra of plasma from patients post radiotherapy, at two months follow up and at eight months follow up. Difference spectra were computed by subtracting the mean spectra of plasma from the patients at post radiotherapy time points from their mean spectra post hormone treatment. The shaded regions depict the spectral regions which are significantly different between each sample set using a two-tailed *t*-test with *p* < 0.001.

**Figure 4 cancers-11-00925-f004:**
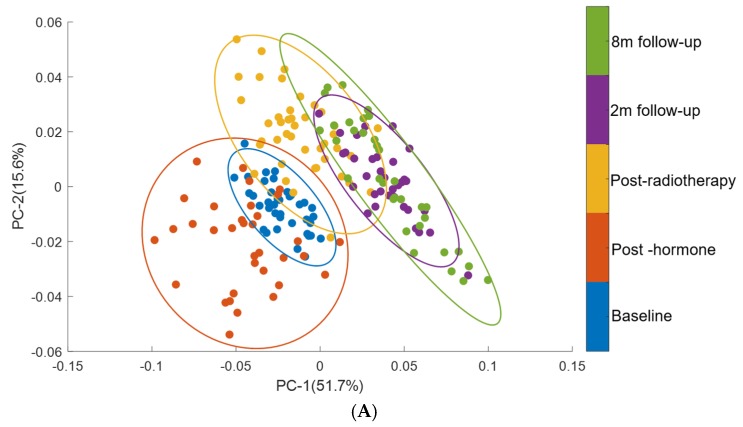
PCA of plasma spectra from patients from baseline through therapy and up to eight months follow up. (**A**) Score plots are shown for patients at baseline (blue), post hormone (red), post radiotherapy (yellow), two months follow up (purple) and eight months follow up (green) (**B**) PC-1 and PC-2 loading plots for regions 1000–1800 cm^−1^ and 2800–3100 cm^−1^. Covariance ellipses (95% confidence) are shown for each class.

**Figure 5 cancers-11-00925-f005:**
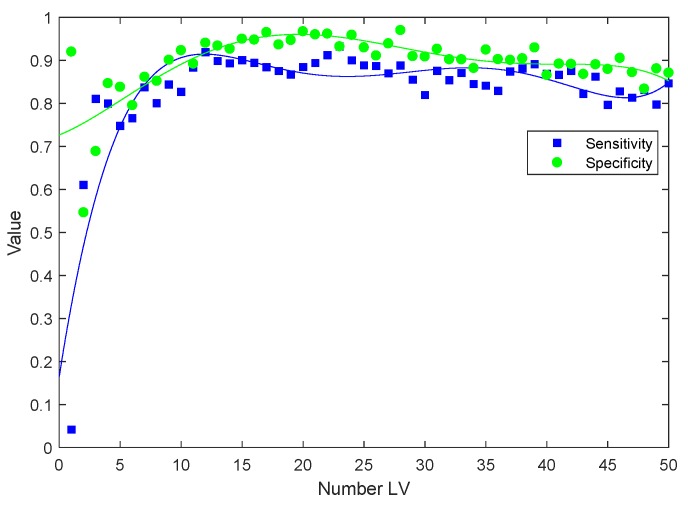
Typical cross-validated sensitivity and specificity for a PLS-DA model developed on FTIR spectra of plasma from patients at baseline and post radiotherapy with increase in the number of latent variables included in the model (LVs).

**Figure 6 cancers-11-00925-f006:**
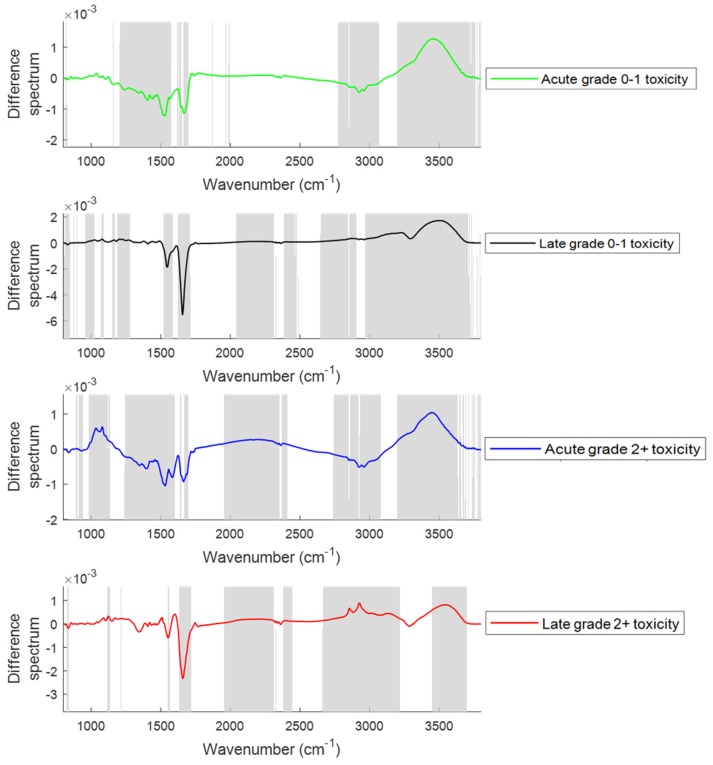
Difference spectra of plasma from patients at baseline and patients suffering from grade 0–1 and grade 2+ acute (immediately after completion of radiotherapy) and late (at eight months post radiotherapy) toxicity. The shaded regions depict the spectral regions which are significantly different between each sample set using a two-tailed *t*-test with *p* < 0.001.

**Figure 7 cancers-11-00925-f007:**
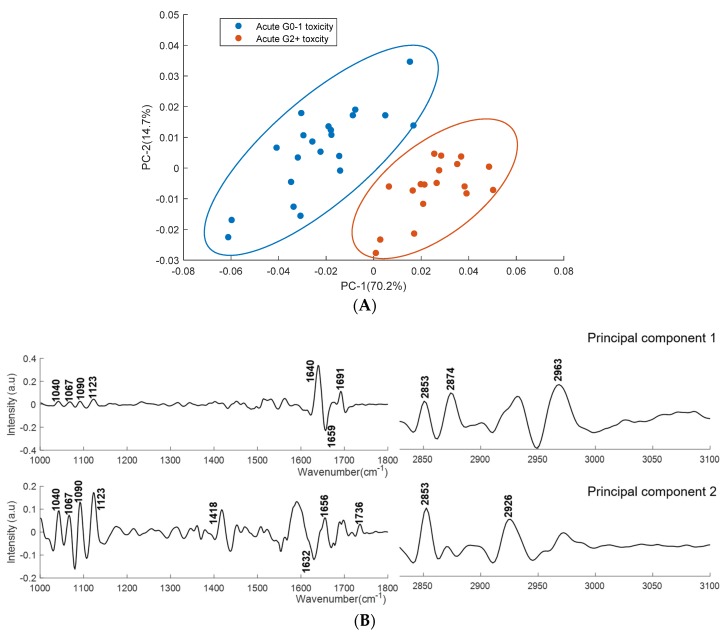
PCA for plasma spectra from patients showing acute grade 0–1 and acute grade 2+ toxicity. (**A**) Score plots are shown for patients showing acute grade 0–1 (blue) and acute grade 2+ toxicity (red). (**B**) PC-1 and PC-2 loading plots for regions 1000–1800 cm^−1^ and 2800–3100 cm^−1^. Covariance ellipses (95% confidence) are shown for each class.

**Figure 8 cancers-11-00925-f008:**
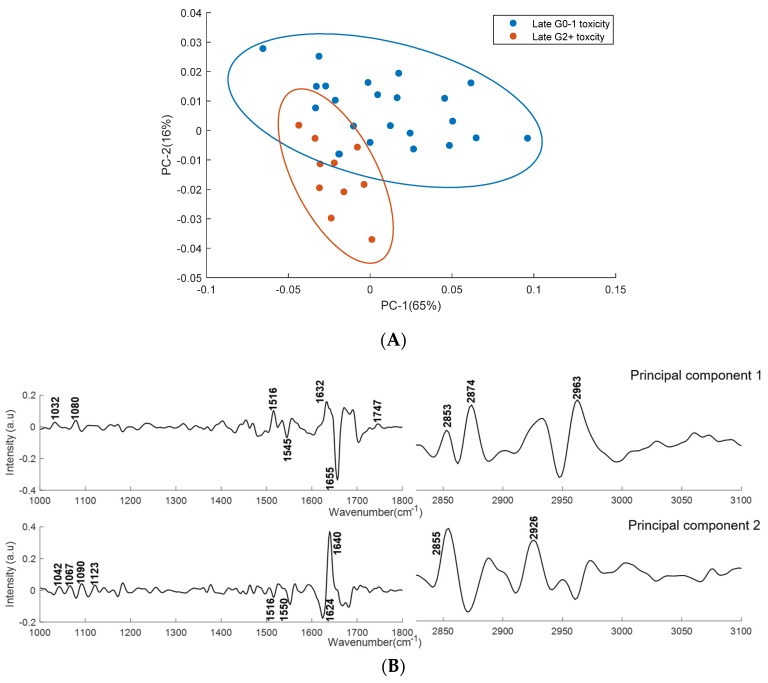
PCA for plasma spectra from patients showing late grade 0−1 and late grade 2+ toxicity. (**A**) Score plots are shown for patients showing late grade 0−1 (blue) and late grade 2+ (red) toxicity. (**B**) PC-1 and PC-2 loading plots for regions 1000–1800 cm^−1^ and 2800–3100 cm^−1^. Covariance ellipses (95% confidence) are shown for each class.

**Figure 9 cancers-11-00925-f009:**
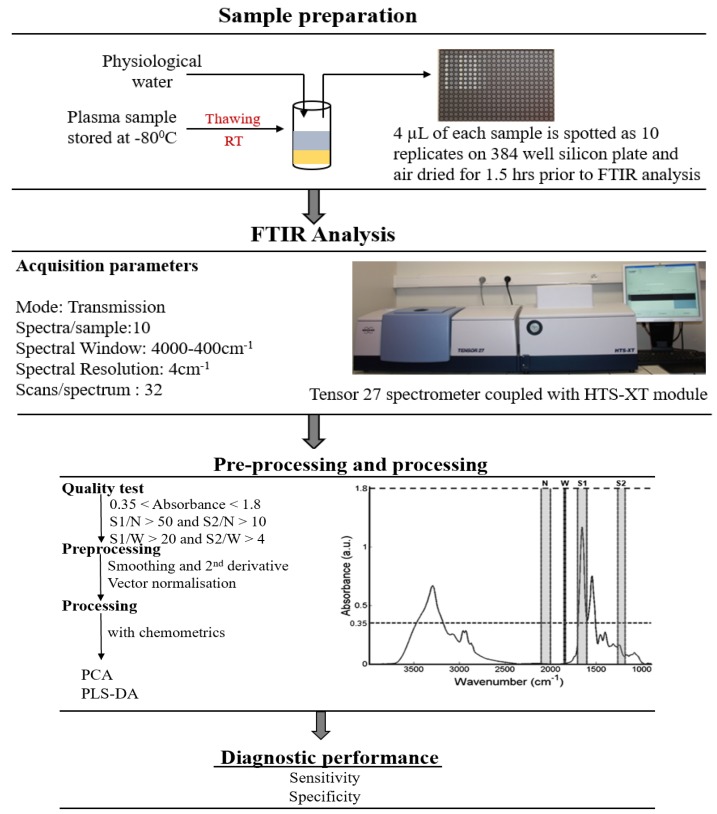
Experimental protocol of high throughput (HT)-FTIR analysis of blood plasma samples from sample preparation to data analysis (reproduced from [23] with permission from The Royal Society of Chemistry). 3-fold dilution: dilution by volume of one part plasma to two parts physiological water.

**Table 1 cancers-11-00925-t001:** Sensitivities and specificities for the PLS-DA classification of patients at baseline versus different treatment stages.

Time Point	Number of Latent Variables (LVs)	Sensitivity	Specificity
Post hormone therapy	10	78.7%	80%
Post radiotherapy	11	89.3%	89.1%
2 months follow up	10	91.4%	91.5%
8 months follow up	13	98.4%	99.1%

**Table 2 cancers-11-00925-t002:** Number of patients showing acute and late toxicity immediately following completion of radiotherapy and at 8 months post radiotherapy.

Toxicity	Number of Patients
Acute grade 0−1 toxicity	24
Acute grade 2+ toxicity	19
Late grade 0−1 toxicity	24
Late grade 2+ toxicity	11

**Table 3 cancers-11-00925-t003:** Sensitivities and specificities for the PLS-DA classification of patients showing acute and late toxicity.

Patients	Number of LVs	Sensitivity	Specificity
Grade 0−1 vs Grade 2+ acute toxicity	10	80.8%	81.6%
Grade 0−1 vs Grade 2+ late toxicity	10	81.4%	81.5%

**Table 4 cancers-11-00925-t004:** Prostate specific antigen (PSA) levels at baseline and at different treatment stages.

Patients	PSA Level Mean (SD)
Baseline (*n* = 37)	14.4 (13.9) ng/mL
Post hormone (*n* = 36)	0.72 (1.71) ng/mL
Post radiotherapy (*n* = 43)	0.08 (0.11) ng/mL
2 months post RT (*n* = 37)	0.07 (0.09) ng/mL
8 months post RT (*n* = 35)	0.09 (0.11) ng/mL

**Table 5 cancers-11-00925-t005:** Clinical features of prostate cancer patients used in this study.

Age (years)	
Mean	69.26
Median	70.5
Range	57−85
PSA (ng/mL)	
Mean	17.22
Median	9.4
T Stage	
T2a to T2c	11 (25%)
T3a	23 (53%)
T3b	08 (18%)
T4a	01 (2%)
Gleason score	
7	14 (33%)
8	16 (37%)
9	13 (30%)
Planned duration of hormones	
6 months	05 (12%)
36 months	38 (88%)
RT Dose/fractions	
81.0/45	43 (100%)

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
