# Peer review of "Monitoring Radiotherapeutic Response in Prostate Cancer Patients Using High Throughput FTIR Spectroscopy of Liquid Biopsies"

_cancers, 2019, doi:10.3390/cancers11070925_

Reviewer 1 Report

The paper describes a promising new approach to develop a biomarker for radiation toxicity, that of FTIR spectroscopy. The aim of developing such a biomarker is an important one for radiotherapy research, because it would allow personalisation of treatment to reduce toxicity. The approach is novel for the field and the paper describes some interesting and useful data.

I have some minor issues and questions:

The figure legends throughout need better descriptions to explain the content. For example Fig 5 legend says it shows the difference between patients at baseline, but it is not explained against what. I assumed it the difference from baseline to 9 months post-radiotherapy, but that needs to be easier to understand (in the body text as well).

Terms like “second derivative mean spectra” need to be explained.

 Most of the comparisons are between baseline and various time-points afterwards, but it would be interesting to know about the differences from other points. For example patients may enter an observational radiotherapy study after hormone treatment, so it would be useful to understand the differences from the post-hormone to the post-RTx samples.

Radiotherapy toxicity is presented as a single end-point but actually the biological basis of GI and GU toxicity in prostate cancer patients is somewhat different. I suggest a separate analysis comparing patients with and without GI and GU toxicity separately. Particularly for GU toxicity it is important that the pre-treatment symptoms are taken into account, either by considering the change in symptoms or using a regression approach.

The analyses of predictive value for toxicity appear to use the difference in FTIR profiles from baseline (unless I have misunderstood – see point 1). To use this assay as an effective radiotherapy predictive test would require only the baseline (or post-hormone) data, and comparing that between patients with and without toxicity. If possible it would be good to see that analysis here.

I am not an expert in spectroscopy studies, but in conventional biomarker studies I expect to see estimations of the statistical significance of differences. Is it not possible to generate p values for the difference spectra? Perhaps by splitting the profiles into bins of arbitrary width and using ANOVA? Without that it is hard to know which peaks and troughs represent genuine differences and which are chance variation. It would also allow correction for multiple testing.

Author Response

Dear Reviewer,

We would like to thank you for your valuable comments. We have addressed all the queries in the manuscript and highlighted the responses in the attached document.

We would be happy to provide any further information if required.

Regards,

Dinesh

Reviewer 2 Report

The authors presents interesting novel findings of FTIR spectroscopy for the detection of RT toxicity using liquid plasma biopsies in prostate cancer patients undergoing RT and ADT. 

The study results are limited by the small study cohort (n=43), a larger sample size would be require to validate the findings. All patients in the study received ADT and there was no RT alone group, hence the changes observed could be attributed to the duration of ADT as opposed to RT.

RT toxicity grading would be more clinically meaningful represented as grade 0-2 vs grade 3 - 4, as grade 2 acute toxicity is expected during RT.  This would likely change the significance of their findings. The authors did not distinguish between GI and GU toxicity.

The article would benefit from further details regarding data analysis methods in the methods section.

The results of the study would be strengthen by the addition of supporting data showing protein, lipid, fatty acid or circulating DNA levels by confirmatory biological assays. Should be considered in future research efforts.

Author Response

(The authors gave the same response as above.)
